# Equity incentive contract characteristics and company operational performance—An empirical study of Chinese listed companies

**Mingzhe Qiao[1], Saihong Chen[1], Shiwei Xu [2,3]***

**1** School of Finance & Management, Shanghai University of International Business and Economics, Shanghai, China, **2** College of Business, Shanghai University of Finance and Economics, Shanghai, China, **3** School of Economics and Management, Shanghai Ocean University, Shanghai, China

* swxu@shou.edu.cn

## Abstract

Equity incentive, as an institutional arrangement for the coordination of the interests of shareholders and managers, has been widely implemented by public companies in developed capital markets throughout Europe and America. However, does it work and/or when might it be more effective in emerging market economies such as China? We aimed to understand the effects of equity incentive plans implemented by listed companies in China and the potential influence of the general characteristics of contracts on the effectiveness of equity incentive plans. Based on behavioral decision theory, this paper adopts a multivariate linear regression model to analyze the 1695 equity incentive plans implemented in Chinese listed companies between 2010 and 2018 with their two-year lagged performance data. The empirical results show that the operational performance of companies after implementing equity incentive plans shows a trend of polarization. In the 95% confidence interval, the effect of restrictive stock incentive and exercise-constrained variables is not significant, while the validity period has a significant positive correlation and incentive intensity has a significantly negative correlation with the company's operational performance. Furthermore, the negative effects mentioned above become more obvious with a longer plan implementation period. Based on these conclusions, we suggest that companies could adopt equity incentive plans with a relatively longer validity period and more reasonable incentive intensity. Additionally, it would be better for companies to select non-restricted stocks as incentive tools if there is no obvious preference.

## 1. Introduction

Modern enterprises incentivize their human capital by issuing stocks to their top managers and/or most important employees and expect to achieve greater expansion and enhance their core competitiveness in the industry. Consequently, the ownership of modern listed companies is scattered and they are controlled by a large number of investors. Meanwhile, Chinese stock investors often consider speculation the main goal, and, currently, in many situations,

**Data Availability Statement:** Sample data could be accessed from the Wind Database (from sub-database "Implementation details of equity incentive" through Wind Client) The website of

Wind Database: "https://www.wind.com.cn/". The authors did not have any special access privileges that other users would not have.

**Funding:** This paper is supported by the Key Program of National Social Science Foundation of China (Grant No. 20AJY008); Shanghai Philosophy and Social Sciences Project (Grant No. 2021BGL009). The funders had no role in study design, data collection and analysis, decision to publish, or preparation of the manuscript.

**Competing interests:** The authors have declared that no competing interests exist.

the benefits that investors obtain do not correspond to their management participation efforts. Moreover, even major shareholders lack relatively professional operational ability and management experience. Therefore, modern joint-stock limited enterprises often employ professional managers (agents) to manage their businesses. Due to their different utility functions, the separation of the company's ownership and management results in an "agency problem". Based on "Self-Interested Behavior Principles", management always seeks to maximize its own interests and, therefore, generates opportunistic behaviors.

Therefore, to coordinate the interests of principals and agents, reduce executive opportunism, alleviate the trust crisis between executives and shareholders, and resolve other issues brought about by the separation of ownership and management, some modern enterprises are attempting to improve the executive compensation system. "Herzberg's motivation–hygiene theory" suggests that a basic salary can only serve as a hygiene factor in the salary system, and not as a motivator. Therefore, the equity incentive plan has been widely accepted and introduced in corporate governance owing to its special properties; for example, it encourages management to reach certain performance standards before implementation and assimilates the interests of principals and agents after implementation [1].

China began implementing real equity incentive plans in 2006, later than some foreign countries. On January 1, 2006, the China Securities Regulatory Commission formally promulgated the "Measures on Administration of Equity Incentive for Listed Companies". Subsequently, equity incentives have officially become part of the salary system of listed companies in China and they have been entrusted with the mission of solving the principal–agent problem of enterprises. However, owing to the relatively short practice time, the immature legal system of Chinese securities, and corporate governance practices, the effectiveness and mechanism of the equity incentive plan are still under-explored [2]. It has been argued that the equity incentive mechanism works as a form of welfare or a motivator. Past literature has determined the validity or invalidity of the equity incentive mechanism from various perspectives, combined with empirical analysis. For example, research finds that equity incentives are negatively correlated to the stock price crash risk in the golden hand cuff regions, while equity incentives are positively correlated to the stock price crash risk in the golden watch region [3]. In addition, research reveals that high equity incentives could strengthen CEOs' degree of earnings management activities [4]. Some studies have analyzed the inner mechanisms of effective share incentive plans from the perspective of property rights, and the results show that state-owned companies' share prices contain large information noise, while non-state-owned companies do not consider stock information content adequately during the process of designing equity incentive plans; hence, their equity incentive plans would lead to inevitable failure in the above-mentioned circumstances [5]. Other scholars have compared the heterogeneity of equity incentive plans between high-tech enterprises and other enterprises, showing that there is a significant conflict between ownership and management in high-tech enterprises; hence, the higher the proportion of stocks held by the largest shareholder, the worse the equity effect [6]. However, the supervisory role between shareholders and management in non-high-tech companies is not obvious [6]. Therefore, it can be found that many exogenous variables, such as property ownership and industry, have a certain effect on the performance of equity incentive plans. Compared with developed capital markets such as Europe, America, and Japan, China's equity incentive system was implemented late, the system design is not perfect, and the equity incentive schemes implemented by listed companies are mixed.

In an emerging economy with an immature capital market and weak corporate governance, how can listed companies design their equity incentive plans so as to contribute to improving their operational performance? Answering this question will help to further clarify the

economic effect of equity incentive, so as to provide more specific reference value for public firms in emerging market countries.

Therefore, this paper focuses on the impact of design factors such as the equity incentive mode, effective period, and other endogenous conditions on the performance of plan implementation. The results of a data analysis of 1695 equity incentive cases among China's listed companies indicate that equity incentive plans are more popular in high-tech industries. The validity periods of incentive plans have a significant positive impact on firms' operational performance, while incentive intensity has a significant negative effect on firms' operation performance. However, exercise constraints of incentive plans have no significant impact on firms' operational performance. Our findings enrich the literature on equity incentive and provide some practical implications for the design of equity incentive plans.

## 2. Literature review

### 2.1. Goal of equity incentive

Existing studies indicate that there are two types of interactions between shareholders and management: the convergence of interest effect [7] and the entrenchment effect [8, 9]. The goal of the equity incentive plan is to use a conciliatory policy to enhance supervision, maintain the convergence of interests between shareholders and managers, and simultaneously reduce conflicts. The equity incentive plans of listed companies play their role mainly in three aspects. First, employee responsibilities are emphasized, and potential profits are shared with employees. According to the optimal contract theory, profit and risk sharing between shareholders and management prompts managers to eliminate short-term behavior and make decisions in accordance with the principle of maximizing shareholder value. Therefore, equity incentive plans can effectively alleviate the agency conflict between shareholders and managers and reduce agency costs [10, 11]. For example, related research points out that cooperation between managers and outside owners is smoother due to equity incentive plans [12].

Second, an equity incentive plan should reduce a company's human capital supervision cost. Previous research has found that equity incentives help to coordinate the interests of shareholders and managers; additionally, they are an effective alternative supervision mechanism in uncertain environments [13]. By granting stocks, corporate administrators and owners could achieve the convergence of interests, and, therefore, trust crises and moral hazards are reduced. Finally, core talent is reserved by granting equity. Studies have revealed that the implementation of an equity incentive plan indicates that a company attaches great importance to human capital and is willing to optimize their corporate governance [14–16]. This approach improves the enthusiasm of core talent, has an important impact on the static and dynamic levels of the capital structure, and finally increases the company's profit.

### 2.2. The design of the equity incentive scheme

The development of the European and American capital markets has a long history, so research on equity incentive has also been conducted earlier. Research discusses the difference between the restricted stock mode and the stock option mode, showing that the former can play a more important role in encouraging senior management when their behaviors influence the company's average outcome [17]. Later, research suggested that the stock option mode is applied more widely than the restricted stock mode, and it is considered one of the best incentive patterns, owing to its active influence in promoting company performance [18]. Homogeneously, additional research reached a similar conclusion by discussing small-scale companies, pointing out that a restricted stock mode could weaken agency behavior and

reduce agency costs [19]. Meanwhile, other research compared the different degrees of effectiveness among some incentive conditions [20].

Chinese scholars have also examined the relationships between the design characteristics of equity incentive schemes and company performance. Some scholars found that executives with high power have the ability to affect or even decide their own salary, and they can easily formulate a stock option incentive plan that has the characteristics of a low exercise price, nominal right condition, and short incentive period [21]. Research showed that the proportion of large shareholders is negatively related to the proportion of stock options awarded to CEOs in new economic companies [14]. According to the final implementation effect, the equity incentive plan can be classified as a "motivator plan" or a "welfare plan". After controlling other variables, research found that the cumulative growth rate of companies that had implemented a "welfare plan" was significantly lower than that of those that had implemented a "motivator plan" [22]. Therefore, it could be concluded that the design of the equity incentive plan determines the motivator or welfare nature of the plan, consequently determining the effect of the incentive plan, thereby affecting the company's performance.

## 2.3. Impact of equity incentive

Mainly, there are two opinions concerning the effect of equity incentive plans: "Optimal Contract Theory" [23, 24] and "Managerial Power Theory" [25]. "Optimal Contract Theory" holds that the equity incentive plan can facilitate the convergence of the core interests of management and shareholders better, thus achieving the purpose of reducing agency costs to solve the "agency problem" of enterprises effectively. For example, research revealed that equity incentive exerts a positive influence on an enterprise's innovation [26]. On the contrary, "Managerial Power Theory" holds that, owing to the existence of information asymmetry and "adverse selection" and "moral hazard" problems, the equity incentive plan is far from perfect. According to this theory, executive compensation is not only an imperfect potential tool for solving the "agency problem," but also a part of the "agency problem" itself [27]. Based on these two theories, Chinese and Western scholars have conducted many empirical explorations. Researchers conducted an empirical study of 202 medium- to large-sized enterprises in Ukraine, and the results showed that the equity incentive plan effectively improved the performance of enterprises during the 1998–2000 period, and that equity incentive plans have a significant positive correlation with corporate performance [28]. An empirical study of 1773 companies in 1990–1996 revealed that the operational profit and cash flow growth rate due to the implementation of the equity incentive policy of the company were significantly higher than those of other companies [29]. Furthermore, other research revealed that equity incentive could exert a significant influence on corporate innovation performance [30]. Therefore, this is in line with the opinion that equity incentives can improve a company's performance. However, some experts oppose this view. Researchers studied the effectiveness of the equity incentive plans of 87 European listed companies and found no significant correlation between equity incentive plans and company performance [31]. Additional research studied the data of Danish listed companies during the 1998–2001 period and found that, as executives can determine salaries themselves, equity incentive plans are inefficient for them [32].

Similarly, domestic scholars' research on the implementation effect of equity incentive plans in China is inconsistent. Researchers selected 59 Chinese listed companies that used equity incentive plans and showed that equity incentives could improve companies' performance; however, the effect differed between share incentives and option incentives, because the latter plan may alleviate agency problems more effectively [33]. However, some scholars hold the view that equity incentive plans cannot effectively solve the "principal–agent

problem". Chinese corporate governance has made great progress recently; however, the effect of the Western equity incentive theory in Chinese conditions is unclear. Hence, researchers studied Chinese listed companies and revealed that equity incentives may increase managers' fraud tendency, which could influence companies' long-term performance, and this phenomenon is especially true for state-owned companies [2]. Based on this, we find that the equity incentive plan itself is not perfect, and various types of endogenous/exogenous conditions, such as the design of each element and the market position of the company, affect its influence to a certain extent. Therefore, differences in the selection of samples and variables will lead to differences in the conclusions.

## 3. Research hypotheses

Most of the existing literature shows optimism about the effectiveness of equity incentive plans for listed companies in Shanghai and Shenzhen. Using Chinese listed companies, including Shanghai and Shenzhen's main boards and small and medium-sized boards, and the Growth Enterprise Board, researchers compared corporate performance with equity incentive plans and without equity incentive plans, and they found that the former was better than the latter [34]. Similarly, researchers focused on listed companies in Shanghai and Shenzhen from 2007 to 2016 and showed that the more managers held the share percentage, the more likely they were to engage in positive strategic response innovation, which may enhance companies' sustainable development capacity [35]. As a normal corporate governance mechanism, equity incentives usually play an important role in preventing or reducing managers' immoral behavior and attracting and retaining talent [36, 37], and they can help to mitigate agency conflicts and promote corporate performance [10]. Meanwhile, the heterogeneity of equity incentive contract terms can lead to different governance effects; this has attracted the attention of some researchers [38, 39]. Given this, from the inner perspective of equity incentive contracts, this paper summarizes and identifies four appropriate contract characteristic variables (incentive subject matter, validity, incentive intensity, and exercise restriction) for further research.

### 3.1. Equity incentive mode and company's operational performance

Stock options, restricted stock, and stock appreciation rights are three common approaches to equity incentive plans. In some cases, companies also use compound incentive approaches to achieve their goals. However, all incentive arrangements are intended to share profits and risks [40]. Nevertheless, different incentive approaches have different effects. Researchers pointed out that equity incentive plans could increase corporate R&D investment and bring better development opportunities; in contrast, restricted stock awards could hamper corporate R&D development [41]. Currently, many American companies implement equity incentive plans; however, scholars cannot pay more attention to the discussion of this theory [1]. Some studies have revealed that under the circumstances of restricted stock awards, managers worry about their financial benefits being negatively affected, so they prefer to adopt a conservative strategy in order to avoid risk. Therefore, a poor decision that lacks efficiency could lead to lower corporate performance [42]. Based on the research mentioned above, this paper proposes the following assumption:

*Hypothesis 1: Compared with restricted stock awards, other equity incentive modes have a significantly positive influence on corporate performance.*

## 3.2. Validity period and company's operational performance

The China Securities Regulatory Commission stipulates that the validity period of equity incentives should generally be no less than one year and no more than 10 years. A short validity period will stimulate short-term agency behavior and cause managers to favor the cashable program in the short term; in this sense, R&D activities are unlikely to become managers' dominant strategy owing to the long period of R&D activities. On the contrary, managers will maintain high conformity with the company if there is a long incentive period driven by common interests, and top managers have more impetus to improve companies' long-term competitiveness. Studies have revealed that the duration of equity incentives is positively related to company performance. Based on the points mentioned above, this paper proposes the following assumption:

*Hypothesis 2*: *There is a significant positive correlation between the validity period and corporate operational performance.*

## 3.3. Incentive intensity and company's operational performance

Generally, incentive intensity is measured by the proportion of shares or holding value [40]. Companies currently adopt different incentive intensities. Some scholars point out that the stronger the perception of motivation, the more significant its effect; in other words, weak incentive intensity cannot play a suitable role in achieving the incentive's aim [40]. For example, if the manager holds many shares in the company, he or she will pay a great opportunity cost when sacrificing the company's profit to satisfy his or her own self-interest; hence, both the principal and agent achieve remarkably consistent benefits [10]. On the other hand, too high an incentive intensity will enhance the manager's power and result in inert behavior; further, the "insider control" phenomenon could become more serious [25]. However, there is a contradiction in the relationship between equity incentive intensity and business performance. Therefore, this paper proposes the following competitive assumptions:

*Hypothesis 3.a*: *There is a significant positive correlation between equity incentive intensity and corporate operational performance.*

*Hypothesis 3.b*: *There is a significant negative correlation between equity incentive intensity and corporate operational performance.*

## 3.4. Exercise restriction and company's operational performance

Listed companies often include industry performance standards in equity incentive plans to achieve the goal of constraint. As a necessary but insufficient condition of the equity incentive scheme, it can be expected that a plan equipped with exercise restrictions has a greater impact on corporate performance than do other non-restriction plans [40]. A company equipped with high exercise restrictions could arouse the manager's enthusiasm, encourage him or her to finish the accepted task positively, and achieve a win–win situation regarding its own goals and those of the manager [43]. In addition, a plan equipped with exercise restrictions can help to retain excellent managers and attract competent or low-risk-averse chief executive officers [44]. The consequences mentioned above positively affect corporate performance and the sharing of wealth.

*Hypothesis 4*: *There is a significant positive correlation between exercise restrictions and corporate operational performance.*

## 4. Methodology

### 4.1. Sample and sources of data

Because the financial crisis that occurred in 2008 affected the implementation of equity incentive plans to a certain extent and reduced listed companies' performance in the Shanghai and Shenzhen A-share markets, we chose equity incentive plans implemented after 2010 as the research sample. Moreover, most companies have not yet disclosed their 2017 annual report data, and this work requires the financial data of three years; hence, we selected plans implemented in the Shanghai and Shenzhen A-share markets between January 1, 2010 and December 31, 2018. Sample data were collected from the Wind Database (accessed from sub-database "Implementation details of equity incentive" through Wind Client. The website of Wind Database: "https://www.wind.com.cn/".). Thus, we obtained 1737 equity incentive cases.

In order to ensure the accuracy and validity of the research results further, we also screened the samples and selected them in accordance with the following principles:

1. Because the different business content of finance and insurance industries leads to different compositions of company assets, liabilities, and owner's equity, they have an independent accounting system. Therefore, we excluded financial and insurance companies.

2. We excluded invalid values that were missing significant data.

   Following the above steps, 1695 valid samples were obtained.

### 4.2. Variable definition

**4.2.1. Dependent variables.** Unlike the western capital market, the capital market in China is still in its infancy. Institutional investors manipulate stock prices significantly, and investors in the exchange often consider speculation their ultimate goal, rather than investment. High-frequency trading, insider information, and asymmetric information greatly influence stock price fluctuations. Therefore, Tobin's Q value, which is widely adopted in research on developed capital markets, is unsuitable for use [33, 45]. Instead, Chinese scholars often use the return on total assets (ROA) in similar studies to measure companies' operational performance. For example, researchers explored the impact of equity incentive schemes on company operational performance using the rate of return on equity (ROE), the rate of return on total assets (ROA), and the main business profit (ROM) as dependent variables [45]. Based on the abovementioned studies, we selected the return on total assets (ROA) as the dependent variable.

**4.2.2. Independent variables.** To test the above hypothesis, we applied the equity incentive approach (STOCK), validity period (PERI), incentive intensity (PPS), and exercise constraint (PECI) as independent variables. Among them, the validity period (PERI) and incentive intensity (PPS) are continuous variables, while equity incentive subject matter (STOCK) and exercise constraint (PECI) are virtual variables. Because the stock option, restricted stock, and stock appreciation rights are the three most common subjects of equity incentive plans and we found that more than half of the companies adopt restricted stock as the incentive subject matter, we set STOCK to 1 if the company used the restricted stock incentive mode; otherwise, it was set to 0. Similarly, we set PECI to 1 if the equity incentive plan set specific exercise constraints; otherwise, it was set to 0. Following common practice, the incentive intensity variable was measured as a percentage of the total equity by the equity incentive plan.

**4.2.3. Control variables.** To control for the impact of internal specific factors on the performance of a company, we included several control variables in the regression models. In

**Table 1. Definitions of variables.**

| Category | Variables | Symbol | Definitions |
|---|---|---|---|
| Dependent Variables | Return on total assets | ROA$_{ave}$ | The mean value of return on total assets for three years after implementation (t to t+2) |
| Independent Variables | Equity incentive approach | STOCK | Restrictive stock incentive mode: Yes, STOCK = 1 |
| | | | No, STOCK = 0 |
| | Validity period | PERI | Actual validity period |
| | Incentive intensity | PPS | Granted shares of equity incentive plan / Total circulation shares of the company |
| | Exercise constraint | PECI | Has set up specific performance standards, Yes, PECI = 1 |
| | | | No, PECI = 0 |
| Control Variables | Size | SIZE | The natural logarithm of the total assets |
| | Leverage | LEV | Total liability / Total assets |
| | Ownership concentration | TOP10_share | Total share proportion of top ten shareholders |
| | Assets turnover rate | ATO | Operating income / Average total assets |
| | Growth of net profit | GROW | Net profit growth (%) = (Current period NP–Prior period NP) / Prior period NP |
| | State ownership | STATE | The company is owned by the state: |
| | | | Yes, STATE = 1; No, STATE = 0 |
| | Year | YEAR | N year data, set n-1 virtual variables |
| | industry | IND | N industry, set n-1 industry virtual variables |

terms of index selection, this paper draws on the control variables of some previous studies [33, 34, 45–47]. Therefore, we selected company size (SIZE), leverage (LEV), equity concentration (Top10_share), total asset turnover (ATO), and net profit growth rate (GROW) as the main control variables. In addition, to control for the influence of year and industry on business performance, we set year (YEAR) and industry (IND) as control variables. All these variables and their definitions are listed in Table 1.

## 4.3. Model specification

To test the hypotheses, we constructed the following multivariate linear regression models to explore the effect of each contract characteristic variable on operational performance:

Formula (1) is the base model, which only includes control variables.

$$\text{ROA}_{ave} = \alpha + \beta_1 * SIZE + \beta_2 * LEV + \beta_3 * TOP10\_share + \beta_4 * ATO + \beta_5 * GROW + \beta_6 * STATE + \beta_7 * YEAR + \beta_8 * IND + \varepsilon \tag{1}$$

Hypotheses 1 to 4 were tested by formula (2) as follows, which includes the independent variables.

$$ROA_{ave} = \alpha + \beta_1 * STOCK + \beta_2 * PERI + \beta_3 * PPS + \beta_4 * PECI + \beta_5 * SIZE + \beta_6 * LEV + \beta_7 * TOP10\_share + \beta_8 * ATO + \beta_9 * GROW + \beta_{10} * STATE + \beta_{11} * YEAR + \beta_{12} * IND + \varepsilon \tag{2}$$

We further specified formula (3) to analyze the effect of the independent variables on the ROA of specific individual years from $t$ to $t + 2$.

$$\text{ROA}_i = \alpha + \beta_1 * STOCK + \beta_2 * PERI + \beta_3 * PPS + \beta_4 * PECI + \beta_5 * SIZE + \beta_6 * LEV + \beta_7 * TOP10\_share + \beta_8 * ATO + \beta_9 * GROW + \beta_{10} * STATE + \beta_{11} * YEAR + \beta_{12} * IND + \varepsilon \tag{3}$$

In this equation, $i$ represents each year from $t$ to $t + 2$.

To test the robustness of the results of the regressions, we chose Tobin's Q as an alternative variable to evaluate the operational performance of a company. Similarly, we constructed multivariate linear regression models for the robustness check as follows.

Formula (4) is the base model; it includes only control variables.

$$
\begin{aligned}
TobinQ_{ave} = \alpha + \beta_1 * SIZE + \beta_2 * LEV + \beta_3 * TOP10\_share + \beta_4 * ATO + \beta_5 * GROW + \beta_6 \\
* STATE + \beta_7 * YEAR + \beta_8 * IND + \varepsilon
\end{aligned}
\tag{4}
$$

Formula (5) is specified to check the robustness of the results of the regressions, which is based on formula (2).

$$
\begin{aligned}
TobinQ_{ave} = \alpha + \beta_1 * STOCK + \beta_2 * PERI + \beta_3 * PPS + \beta_4 * PECI + \beta_5 * SIZE + \beta_6 * LEV \\
+ \beta_7 * TOP10\_share + \beta_8 * ATO + \beta_9 * GROW + \beta_{10} * STATE + \beta_{11} * YEAR \\
+ \beta_{12} * IND + \varepsilon
\end{aligned}
\tag{5}
$$

Formula (6) is specified to verify the effect of the independent variables on the companies' operational performance in specific individual years from $t$ to $t + 2$.

$$
\begin{aligned}
TobinQ_i = \alpha + \beta_1 * STOCK + \beta_2 * PERI + \beta_3 * PPS + \beta_4 * PECI + \beta_5 * SIZE + \beta_6 * LEV \\
+ \beta_7 * TOP10\_share + \beta_8 * ATO + \beta_9 * GROW + \beta_{10} * STATE + \beta_{11} * YEAR \\
+ \beta_{12} * IND + \varepsilon
\end{aligned}
\tag{6}
$$

In this equation, $i$ represents each year from $t$ to $t + 2$.

## 5. Empirical analysis results

### 5.1. Descriptive analysis

The results of the descriptive analysis of the variables of multiple linear models are shown in the table below (Table 2).

A comparison of the three-year ROA data after implementation shows that the mean value of ROA is declining, but the maximum value is increasing, and it shows a growing polarization trend over the three years. Therefore, the operational performance of listed companies who

**Table 2. Descriptive statistics of variables.**

| VARIABLES | Min | Max | Mean | S D | Skewness | Kurtosis |
|---|---|---|---|---|---|---|
| $ROA_t$ | -57.47 | 44.02 | 8.434 | 6.593 | -1.031 | 19.50 |
| $ROA_{t+1}$ | -56.22 | 53.82 | 7.608 | 7.518 | -1.195 | 17.59 |
| $ROA_{t+2}$ | -78.93 | 67.60 | 6.592 | 9.192 | -1.995 | 18.71 |
| $ROA_{ave}$ | -33.76 | 34.87 | 7.544 | 6.250 | -0.510 | 7.880 |
| STOCK | 0 | 1 | 0.727 | 0.446 | -1.018 | 2.037 |
| PERI | 0.001 | 9.996 | 2.195 | 1.623 | 1.581 | 6.635 |
| PPS | 1 | 10 | 4.444 | 0.921 | 2.717 | 17.25 |
| PECI | 0 | 1 | 0.998 | 0.0420 | -23.71 | 563.0 |
| SIZE | 19.69 | 27.39 | 21.93 | 1.211 | 1.102 | 4.569 |
| LEV | 2.439 | 106.3 | 37.58 | 18.52 | 0.422 | 2.624 |
| TOP10_share | 15.95 | 96.75 | 61.39 | 13.92 | -0.465 | 2.951 |
| ATO | 0.0104 | 10.81 | 0.715 | 0.524 | 6.334 | 94.30 |
| GROW | -3,290 | 13,132 | 75.55 | 606.2 | 16.46 | 336.3 |
| STATE | 0 | 1 | 0.135 | 0.342 | 2.135 | 5.558 |

Note: Valid individual cases (N) = 1695; SD = standard deviation.

have implemented equity incentive plans experienced large-scale variance, which means that the impact of stock incentive plans with different characteristics might be different.

Regarding the design of the equity incentive plan, 72.7% of sample cases are restricted stock incentive plans, which means that the companies prefer to use restricted stocks as incentive tools. The differences in incentive intensity between the companies are large. Most listed companies set up company performance conditions when exercising to ensure the motivational effect of an equity incentive plan to a certain degree, as the mean value of *PECI* is 0.998.

## 5.2. Results of multiple linear regressions

Pearson correlation coefficients for each variable are presented in Table 3. It is worth noting that there was a negative correlation between STOCK and the other variables. Because STOCK is a virtual variable and 1 represents restricted stocks, while 0 represents non-restricted stocks, it can be found that restricted stock has an inhibitory effect on other explanatory and control variables. By comparing all Pearson correlation indices between variables, it can be found that all values of correlation coefficients are below 0.4 (a few indices are around 0.4). Hence, the correlations between these variables are weak. Therefore, there should be no multiple co-linear problems between these variables in the regression models.

Next, we obtained the multiple linear regression result of ROA (the results are shown in Table 4). The data of continuous variables in the regression models were winsorized at the 1% level.

From the results of the regression model shown above, it can be seen that the adjusted coefficients of determination are above 0.25, which means that these models have strong explanatory ability. A comparison of the last four models (Models (6)-(8)) shows that Model (6)'s goodness of fit is the best, Model (7) is the second best, and Model (8) is the worst. The results show that all the regression models are significant at the 0.001 level.

By comparing the coefficients of each independent variable in these eight models, the effect of each equity incentive plan's characteristic variable on the operational performance of a company can be discussed as follows.

**Table 3. Pearson correlation coefficients.**

| | ROA$_{ave}$ | STOCK | PERI | PPS | PECI | SIZE | LEV | Top10_share | ATO | GROW |
|---|---|---|---|---|---|---|---|---|---|---|
| **ROA$_{ave}$** | 1 | | | | | | | | | |
| **STOCK** | 0.021 | 1 | | | | | | | | |
| **PERI** | 0.054* | -0.131** | 1 | | | | | | | |
| **PPS** | -0.096** | -0.138** | -0.041$^{†}$ | 1 | | | | | | |
| **PECI** | -0.047$^{†}$ | -0.026 | 0.051* | 0.043$^{†}$ | 1 | | | | | |
| **SIZE** | -0.057* | -0.106** | 0.164** | -0.062* | 0 | 1 | | | | |
| **LEV** | -0.233** | -0.097** | 0.064** | 0.116** | -0.002 | 0.607** | 1 | | | |
| **TOP10_share** | 0.211** | -0.034 | 0.125** | -0.134** | -0.035 | -0.102** | -0.103** | 1 | | |
| **ATO** | 0.146** | -0.031 | 0.045$^{†}$ | 0.003 | -0.030 | 0.103** | 0.199** | 0.100** | 1 | |
| **GROW** | 0.157** | -0.004 | -0.044$^{†}$ | 0.027 | 0.004 | 0.066** | -0.018 | -0.067** | 0.004 | 1 |
| **STATE** | -0.119** | -0.095** | 0.184** | 0.004 | 0.017 | 0.151** | 0.128** | -0.026 | 0.033 | -0.013 |

Note:

$^{†}$ P<0.1 (two-tailed);

*P < 0.05 (two-tailed);

** P < 0.01 (two-tailed).

**Table 4. Multiple liner regression result of ROA.**

| | $ROA_{ave}$ | $ROA_{ave}$ | $ROA_{ave}$ | $ROA_{ave}$ | $ROA_{ave}$ | $ROA_t$ | $ROA_{t+1}$ | $ROA_{t+2}$ |
|---|---|---|---|---|---|---|---|---|
| | (1) | (2) | (3) | (4) | (5) | (6) | (7) | (8) |
| STOCK | | 0.229 | 0.270 | 0.203 | 0.190 | 0.622* | -0.0870 | 0.243 |
| | | (0.77) | (0.91) | (0.68) | (0.64) | (2.24) | (-0.25) | (0.57) |
| PERI | | | 0.281* | 0.269† | 0.283* | 0.251† | 0.334* | 0.225 |
| | | | (1.96) | (1.88) | (1.97) | (1.89) | (2.01) | (1.11) |
| PPS | | | | -0.184* | -0.177* | -0.183* | -0.160† | -0.113 |
| | | | | (-2.20) | (-2.12) | (-2.36) | (-1.66) | (-0.96) |
| PECI | | | | | -4.560 | -5.826* | -4.323 | -3.639 |
| | | | | | (-1.54) | (-2.12) | (-1.26) | (-0.87) |
| SIZE | 0.662*** | 0.672*** | 0.629*** | 0.576*** | 0.574*** | 0.732*** | 0.588*** | 0.451* |
| | (4.81) | (4.86) | (4.50) | (4.06) | (4.05) | (5.57) | (3.58) | (2.25) |
| LEV | -0.121*** | -0.121*** | -0.120*** | -0.116*** | -0.116*** | -0.124*** | -0.116*** | -0.105*** |
| | (-12.95) | (-12.93) | (-12.85) | (-12.27) | (-12.29) | (-14.12) | (-10.56) | (-7.84) |
| TOP10_share | 0.072*** | 0.073*** | 0.070*** | 0.067*** | 0.067*** | 0.061*** | 0.063*** | 0.068*** |
| | (7.54) | (7.56) | (7.24) | (6.89) | (6.85) | (6.75) | (5.61) | (4.91) |
| ATO | 3.964*** | 3.968*** | 3.954*** | 3.961*** | 3.940*** | 4.180*** | 3.821*** | 3.514*** |
| | (11.48) | (11.49) | (11.45) | (11.49) | (11.42) | (13.07) | (9.58) | (7.21) |
| GROW | 0.006*** | 0.006*** | 0.006*** | 0.006*** | 0.006*** | 0.009*** | 0.006*** | 0.001 |
| | (7.29) | (7.29) | (7.36) | (7.49) | (7.50) | (11.79) | (5.52) | (0.96) |
| STATE | -1.712*** | -1.691*** | -1.812*** | -1.809*** | -1.808*** | -1.075*** | -1.605*** | -2.651*** |
| | (-4.59) | (-4.52) | (-4.78) | (-4.78) | (-4.77) | (-3.06) | (-3.67) | (-4.96) |
| YEAR | Include | Include | Include | Include | Include | Include | Include | Include |
| IND | Include | Include | Include | Include | Include | Include | Include | Include |
| _cons | -7.244* | -7.507* | -7.752* | -5.838† | -1.252 | -6.337 | -4.912 | 4.318 |
| | (-2.16) | (-2.23) | (-2.30) | (-1.68) | (-0.27) | (-1.49) | (-0.93) | (0.67) |
| Adj. $R^2$ | 0.251 | 0.251 | 0.253 | 0.254 | 0.255 | 0.288 | 0.193 | 0.173 |
| F | 19.97*** | 19.34*** | 18.88*** | 18.50*** | 18.04*** | 21.14*** | 12.89*** | 11.44*** |
| N | 1695 | 1695 | 1695 | 1695 | 1695 | 1695 | 1695 | 1695 |

Note:

† $P < 0.1$ (two-tailed);

* $P < 0.05$ (two-tailed);

** $P < 0.01$ (two-tailed);

*** $P < 0.001$ (two-tailed).

t statistics in parentheses.

**5.2.1. STOCK.** STOCK is a virtual variable. Value 1 represents companies choosing restricted stock as an incentive tool and value 0 represents non-restricted stock. The coefficients of STOCK in Table 4 are negative; it can be inferred that the equity incentive scheme with non-restricted stock has a greater incentive effect on a company's operational performance. However, the regression coefficients of STOCK are insignificant (p>0.05), indicating that STOCK has weak explanatory capability. Hence, Hypothesis 1 is rejected.

**5.2.2. PERI.** All the regression coefficients of PERI in Table 4 are positive and significant (p<0.05). Hence, the validity period of the equity incentive plan is positively correlated with ROA. The regression results of Models (6) to (8) indicate that the effect of PERI is strongest in the second year after the implementation of the equity incentive plan. Thus, Hypothesis 2 is supported.

**5.2.3. PPS.** As shown in Table 4, all regression coefficients of PPS are positive and significant ($p < 0.05$ and $p < 0.10$ in Model (4)). Therefore, it can be concluded that incentive intensity has a significant negative influence on a company's operational performance. In addition, the increasing absolute value of the regression coefficient in Models (6) to (8) indicates that there is an decreasing impact of incentive intensity over time. Therefore, when all control variables remain unchanged, a possible explanation might be that high incentive intensity may induce managers' inert behavior, which discourages their contributions to improve companies' performance. However, one year later, stock options awarded to managers become effective to exercise, and an executive would make a greater effort to improve the operational performance of their company for the sake of their own benefit, generated by exercising stock options. Hence, Hypothesis 3.a is rejected, while Hypothesis 3.b is supported.

**5.2.4. PECI.** Owing to relevant policy constraints and incentive purposes, more than 99% of the companies add exercise-constricted conditions in equity incentive plans. The coefficient of variable PECI is not significant ($p > 0.05$). Therefore, Hypothesis 4 is rejected.

The above results indicate that an equity incentive plan with a long validity period can greatly promote a company's operational performance, and the impact reaches its peak in the second year after equity incentive implementation. However, incentive intensity has a negative effect on a company's operational performance.

## 5.3. Robustness check

Table 5 presents the robust check regression results. PERI and PPS maintain significant relationships with a company's operational performance in all the models. Similar to previous research, PERI shows a significant positive correlation, while PPS shows a significant negative correlation with the company's operational performance. However, STOCK is significantly negatively related to Tobin's Q, while the relationship with ROA is not significant. Overall, the results of the robust check are consistent with the previous regressions presented in Table 4.

# 6. Conclusions and further directions

## 6.1. Conclusions and discussion

In this section, we provide an overview of the conclusions drawn from the empirical results. Based on these conclusions, we offer some suggestions on the design of an equity incentive plan for listed companies in Chinese capital markets.

Based on the equity incentive plans implemented from 2010 to 2018, we conducted an in-depth analysis of the impact of equity incentive characteristics on a company's operational performance. The study found that equity incentive plans are widely used in high-tech industries, particularly in the computer software sector. This is because companies in the modern high-tech industry have great potential for rapid development, and the growth of their core talent cannot meet their rising demand. Undoubtedly, senior executives of high-tech enterprises usually have a higher level of professional skills, and such professional skills have a higher threshold. The above factors determine that it is necessary to implement appropriate incentives. In addition, the immature corporate governance of such enterprises and the lack of experience make the "agency problem" extremely acute in high-tech industries.

Meanwhile, the study also revealed that the impact of the equity incentive approach and exercise constraints on operational performance was not significant. This conclusion is different from extant research that found that the "stock option has a more obvious effect on executive incentive". In the mentioned work, the authors used China's GEM listed companies as a sample—this type of listed company emerged late, they have better adaptability and flexibility, and they usually do not experience the inertia of difficult returns [40]. Our results are also

**Table 5. Multiple linear regression result of Tobin's Q.**

|  | $TobinQ_{ave}$ | $TobinQ_{ave}$ | $TobinQ_{ave}$ | $TobinQ_{ave}$ | $TobinQ_{ave}$ | $TobinQ_t$ | $TobinQ_{t+1}$ | $TobinQ_{t+2}$ |
|---|---|---|---|---|---|---|---|---|
|  | (1) | (2) | (3) | (4) | (5) | (6) | (7) | (8) |
| STOCK |  | -0.207* | -0.187* | -0.212*** | -0.210*** | -0.165† | -0.226*** | -0.144 |
|  |  | (-2.55) | (-2.31) | (-2.61) | (-2.58) | (-1.80) | (-2.60) | (-1.48) |
| PERI |  |  | 0.137*** | 0.133*** | 0.130*** | 0.122*** | 0.112*** | 0.122*** |
|  |  |  | (3.51) | (3.41) | (3.34) | (2.79) | (2.68) | (2.61) |
| PPS |  |  |  | -0.068*** | -0.069*** | -0.067*** | -0.058* | -0.078*** |
|  |  |  |  | (-2.98) | (-3.03) | (-2.63) | (-2.39) | (-2.86) |
| PECI |  |  |  |  | 0.836 | 1.194 | 0.725 | 0.364 |
|  |  |  |  |  | (1.04) | (1.31) | (0.84) | (0.38) |
| SIZE | -0.348*** | -0.357*** | -0.378*** | -0.397*** | -0.397*** | -0.494*** | -0.372*** | -0.268*** |
|  | (-9.25) | (-9.46) | (-9.92) | (-10.31) | (-10.30) | (-11.39) | (-9.04) | (-5.83) |
| LEV | -0.028*** | -0.028*** | -0.028*** | -0.027*** | -0.026*** | -0.027*** | -0.027*** | -0.026*** |
|  | (-11.03) | (-11.09) | (-10.97) | (-10.29) | (-10.28) | (-9.36) | (-9.73) | (-8.52) |
| TOP10_share | 0.019*** | 0.018*** | 0.017*** | 0.016*** | 0.016*** | 0.020*** | 0.012*** | 0.016*** |
|  | (7.14) | (7.05) | (6.54) | (6.11) | (6.13) | (6.69) | (4.35) | (5.03) |
| ATO | 0.092 | 0.089 | 0.082 | 0.085 | 0.089 | 0.272*** | 0.056 | -0.070 |
|  | (0.97) | (0.94) | (0.88) | (0.91) | (0.94) | (2.58) | (0.56) | (-0.62) |
| GROW | -0.000 | -0.000 | -0.000 | -0.000 | -0.000 | -0.000 | 0.000 | -0.000 |
|  | (-0.49) | (-0.49) | (-0.37) | (-0.19) | (-0.19) | (-0.42) | (0.54) | (-0.43) |
| STATE | -0.122 | -0.141 | -0.200† | -0.199? | -0.199† | -0.166 | -0.140 | -0.252* |
|  | (-1.19) | (-1.38) | (-1.94) | (-1.93) | (-1.93) | (-1.43) | (-1.28) | (-2.05) |
| YEAR | Included | Included | Included | Included | Included | Included | Included | Included |
| IND | Included | Included | Included | Included | Included | Included | Included | Included |
| _cons | 10.59*** | 10.83*** | 10.71*** | 11.42*** | 10.57*** | 12.55*** | 10.25*** | 8.177*** |
|  | (11.57) | (11.79) | (11.69) | (12.09) | (8.49) | (8.96) | (7.71) | (5.51) |
| Adj. $R^2$ | 0.389 | 0.392 | 0.396 | 0.398 | 0.399 | 0.442 | 0.419 | 0.319 |
| F | 37.02*** | 36.16*** | 35.65*** | 35.01*** | 34.01*** | 40.41*** | 36.86*** | 24.38*** |
| N | 1695 | 1695 | 1695 | 1695 | 1695 | 1695 | 1695 | 1695 |

NOTE:

† $p < 0.10$,

* $p < 0.05$,

*** $p < 0.01$, *** $p < 0.001$. $t$ statistics in parentheses.

different from research indicating that "equity incentive will increase executive speculation and weaken enterprise value". Chinese officials are trying to avoid corporate fraud, but due to the characteristics of emerging markets in the transition period, Chinese enterprises still face the situation of senior executives cheating in the implementation of equity incentive, which is similar to the situation that has occurred in the Western capital markets in the past [2]. There are several possible reasons for this finding. First, due to less practical experience with equity incentives in China, mutual imitation leads to high similarity between the equity incentive plans implemented by listed companies, which might be suitable for each company. Second, the design and implementation of equity incentive plans are constrained by certain laws and regulations, especially for state-owned enterprises that undertake many social functions; due to the senior executives' special status, they cannot implement proper incentive models according to their objective development situation. Undoubtedly, the poor motivating effect is

a consequence. The Chinese economy has experienced rapid growth and development in recent years; however, the system designed for corporate governance lags behind.

Thirdly, the study also found that the validity period was significantly positively correlated with companies' operational performance. This result confirms the findings of some previous studies. Generally, a longer validity period and stricter exercise constraints result in better equity incentive plans [43, 48]. In addition, a long validity period can reduce shortsighted management behavior and indirectly improve a company's performance [48]. The research conclusion is consistent with the views of some previous researchers; for example, some Chinese scholars believe that executives with a relatively short incentive and constraint time may prefer earnings management, stock price manipulation, and other methods to meet the exercise conditions; meanwhile, for a long period of incentive and constraint, senior executives are more likely to truly create enterprise value locally by improving their operating efficiency [40]. Our finding is also similar to the research results on American enterprises; for example, scholars selected 1932 firms as a sample and revealed that long-term incentives have a stronger influence on innovation [30]. However, our results are in contrast to research that found that a longer term of validity will induce excessive short-term investment behavior, which may be related to the term of the manager [49]. Managers have incentives to overinvest in short-term projects to boost the board's attitudes regarding their abilities, especially if the manager faces job changes in the short term, as a longer term of validity incentive will increase their agency behavior.

Lastly, the study suggested that the incentive intensity is significantly negatively correlated with the operational performance of companies, and the negative impact increases with time. There are several possible reasons for this. First, the design of the equity incentive plan is subject to managerial power, which could facilitate management self-interest. Therefore, the greater the managerial power, the lower the positive effect between equity incentive and corporate performance. As is known, the US capital market is more established than that of the UK; hence, US management has a lower level of ownership than its British counterparts, which could strengthen the positive effect between equity incentive and corporate performance [9]. Exorbitant incentive intensity is also a warning for the market to address executives' self-interest behavior. For example, the power held by senior executives can lead to their self-serving behavior. The greater the power held by senior executives, the greater the intensity of equity incentives. Meanwhile, the announcement of a draft equity incentive will produce a significantly positive market response, and the market can be made aware of the self-serving behavior of senior executives. Therefore, incentive intensity is negatively related to market reactions [50], which reflects the results of strong equity incentive and weak operational performance. However, empirical studies on mature US capital markets have arrived at a contrasting conclusion. A study of S&P 500 firms in the 1990s suggested that the market value of the company is improved when the pay for performance plan directly for the company's executives is disclosed; the effect of implementing this plan is obvious in companies with potential agency, because this stock compensation mechanism has stimulated the efforts of executives and led to the improvement of company performance [51, 52]. The next reason is that current equity incentive plans are relatively immature, and evidence of the impact of equity incentive intensity has not yet appeared. On the contrary, the sample of S&P companies in the European and American capital markets is more mature, which means more reasonable policy supervision and market expectations. In addition, this paper only focuses on the effects that occurred during the three years after implementation; under these circumstances, the motivation effect may be lagging and not fully presented yet. The last point is that managers may become less diligent when given high reward equity. Due to the incomplete improvement of the salary system, equity incentive income is viewed as the excess income of management, and it evolves into a

means of safeguarding the management's position. According to Herzberg's two-factor theory, unreasonable equity incentive income plays a role as a hygiene factor rather than a satisfier factor, which would only eliminate managers' dissatisfaction, and it would not increase their job satisfaction or performance [53].

## 6.2. Theoretical contribution and practical implications

The marginal theoretical contribution of the study may be reflected in the following aspects: Firstly, this paper enriches the relevant research framework of equity incentive. Previous studies mostly discussed the economic consequences in terms of single variables, such as the mode, validity period, and intensity of equity incentive. For example, some researchers only selected the percentage of shares held by managers as a proxy for equity incentive [26]. This paper presents a comprehensive study on the above three factors of equity incentive and we used the panel data of Shanghai and Shenzhen's listed companies for the regression test, so as to obtain more prudent and detailed research conclusions. Our work also offers more intuitive empirical evidence to verify whether equity incentive intensifies or alleviates the agency phenomenon. Secondly, this paper broadens the research on the influencing factors of enterprise performance. In the past, many studies have discussed the impact of enterprise financial leverage [47], enterprise risk management [46], corporate social responsibility [54], a Confucian culture [55], the enterprise power structure [56], and other factors on corporate performance. This work explored the impact of equity incentive on corporate value and expanded the research on the prior variables of enterprise performance.

At the same time, this paper may also have some practical implications. Our conclusion shows that the existing equity incentive model does not have a significant impact on corporate performance, which shows that in the practice of enterprise management, we should strengthen the innovation of the incentive model and explore the equity incentive method that is most suitable for Chinese local enterprises, so as to enhance the practical value of the strategy. In addition, this study found that the longer the validity period of equity incentive, the more it can promote enterprise performance. This conclusion shows that the validity period can be appropriately increased when determining the duration of the equity incentive in practice, which would help to maintain the convergence of the interests of management and enterprises. Finally, we found that higher equity incentive intensity is not conducive to enterprise performance. This conclusion indicates that managers holding a higher equity share may be more inclined to engage in self-interested behavior and further contribute to the occurrence of the agency phenomenon. Therefore, in practice, we should determine a reasonable range of equity incentive intensity and carefully select an incentive with reasonable intensity.

## 6.3. Suggestions

Based on the above discussion, the design of an incentive mechanism in a team is a very important issue, which is related to organizational performance [57]. Based on the empirical findings, we offer the following suggestions on the equity incentive plans of listed companies. First, the subject matter shows an unapparent negative correlation with company operational performance. Therefore, enterprises should consider their actual situation first when choosing the incentive medium. If there is no preference for any subject matter, companies could consider non-restricted stocks first. Second, enterprises should create reasonable incentive schemes based on their current situation. Based on the significant effect of the validity period and incentive intensity, enterprises can choose a long incentive validity period and they should be more cautious when selecting the intensity of motivation. Lastly, equity incentive plans can be an efficient means to test the managerial power of a company. Companies with a "welfare

plan" (A type of equity incentive plan that is inclined to reward instead of inspire) should pay more attention to improving the internal management structure and reducing management's power.

## 6.4. Further research directions

This paper discusses the relationship between the types of equity incentive and corporate performance, and it is found that there is no obvious linear relationship between them. In spite of the aforementioned theoretical significance and practical enlightenment, however, there are still some imperfections in this work. In future research, we can further explore the possible U-shaped and other nonlinear relationships between them. In addition, this paper verifies the significant impact of the validity period and intensity of equity incentive on corporate performance, but the mechanism and boundary of equity incentive have not been discussed in depth. In future research, we can further analyze the path mechanism of the impact of both equity incentive validity and incentive intensity on corporate performance, as well as the boundary effects, such as action intensity, in different situations from the perspective of theory and demonstration.

## Author Contributions

**Conceptualization:** Mingzhe Qiao.

**Data curation:** Saihong Chen.

**Methodology:** Mingzhe Qiao, Shiwei Xu.

**Writing – original draft:** Saihong Chen.

**Writing – review & editing:** Shiwei Xu.

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
