## [Decision Letter · Decision Letter 0]

25 Mar 2022

PONE-D-22-03250Equity Incentive Contract Characteristics and Company Operational Performance——An Empirical research of Chinese listed companiesPLOS ONE

Dear Dr. Xu,

Thank you for submitting your manuscript to PLOS ONE. After careful consideration, we feel that it has merit but does not fully meet PLOS ONE’s publication criteria as it currently stands. Therefore, we invite you to submit a revised version of the manuscript that addresses the points raised during the review process.

We look forward to receiving your revised manuscript.

Kind regards,

Haijun Yang, Ph.D

Academic Editor

PLOS ONE

Journal Requirements:

Additional Editor Comments:

Dir Dr. Xu,

The reviewers have submitted the comments with details. Please revise your manuscript according to the comments and polish the language.

Beat,

Yang

Reviewers' comments:

Reviewer's Responses to Questions

**Comments to the Author**

1. Is the manuscript technically sound, and do the data support the conclusions?

Reviewer #1: Partly

Reviewer #2: Yes

2. Has the statistical analysis been performed appropriately and rigorously? 

Reviewer #1: Yes

Reviewer #2: Yes

3. Have the authors made all data underlying the findings in their manuscript fully available?

Reviewer #1: Yes

Reviewer #2: Yes

4. Is the manuscript presented in an intelligible fashion and written in standard English?

Reviewer #1: No

Reviewer #2: Yes

5. Review Comments to the Author

Reviewer #1: The authors study Equity Incentive Contract Characteristics and Company Operational Performance——Empirical research of Chinese listed companies. Here are my comments to the authors that attempt to provide a review and improve the manuscript:

1. Authors should use Plos One style in writing "abstract"; please check the author's instruction.

2. No need to have the results in the introduction section.

3. Authors should add the “research structure” for the whole manuscript in the introduction section.

4. Both Introduction and LR parts need to be improved and enriched with extra new papers in the field.

5. Authors need to position their article compared to the academic literature and specify the gap they are trying to fill.

6. The contribution of this paper and the importance of this contribution are not clear; authors should state their contribution and show the importance of this contribution.

7. Methodology: the data used in this research are from 2010 till 2014; these data are outdated. Updated data are required.

8. The authors are not comparing their results with previous work. They are not referring to any reference.

9. The conclusion should be written in paragraphs, not as points.

10. The results and analysis section; it only contains the results with a little discussion. More discussions are needed. Also, the authors should explain more precisely the effect of their results on the Chinese market. In other words, authors should interpret the results beyond their statistical significance to demonstrate their economic impact.

11. Authors should include the "suggestion" section in the conclusion

12. Authors should add the "practical and theoretical implication" and the "limitations and further studies" sections at the end of their paper.

13. Regarding references: most of the references are old. This topic is considered almost as new topic, and many important articles have been published recently in this field. I advise you to refer to previously published articles in Plos One and other top journals to include more updated papers in your study.

14. Editing for English is required

15. The author mentioned the name of the researchers in addition to the number of the reference in many pages. As in rows of 137, 141, 161, 172, 179, etc

Reviewer #2: Major concerns

The structure of the paper should be presented in the introduction.

The authors should explain why China is worth studying in relation to other countries (e.g., India or Japan).

Despite its drawbacks, TobinQ has been used extensively by papers focused on emerging capital markets (see additional readings). I strongly recommend you to use this measure for robustness checks.

You should clarify the contributions of the paper which are not elaborated well in the current paper. You can talk about the following contributions: What insights can you provide based on your finding? Do they push forward our understanding? What should we do with your research? Do you have any suggestions to improve the current regulation or practice? Adding the above discussion and extending your literature review may help you make more contributions and position your contributions better.

The authors should highlight the limits of their research.

In conclusion, I would like to thank the authors for a very interesting, unique, and potentially important paper. Hope these comments and suggestions can help further their study. I consider that the paper can bring a significant contribution to the extant literature once the above-mentioned recommendations are taken into account.

Additional readings

Leverage and firm growth: an empirical investigation of gazelles from emerging Europe. International Entrepreneurship and Management Journal, 15(1), 209–232. doi: 10.1007/s11365-018-0524-5.

The Impact of Enterprise Risk Management on Firm Value: Empirical Evidence from Romanian Non-Financial Firms, Inzinerine Ekonomika-Engineering Economics, 29(2), 151–157, https://doi.org/10.5755/j01.ee.29.2.16426

6. PLOS authors have the option to publish the peer review history of their article (what does this mean?). If published, this will include your full peer review and any attached files.

Reviewer #1: No

Reviewer #2: No

---

## [Author Response · Author response to Decision Letter 0]

6 Jul 2022

Suggestions of Reviewer 1

Dear expert, we are very grateful for your valuable suggestions, and we will make a specific reply below.

1. Authors should use Plos One style in writing "abstract"; please check the author's instruction.

Response: As suggested, the summary section has been tracked, Please refer to line 13-37 in tracked version.（all “line” refers to tracked version）

“Equity incentive, as an institutional arrangement for coordinating the interests of shareholders and managers, has been widely implemented by public companies in developed capital markets in Europe and America. However, does it work and/or when might it be more effective in emerging market economies like China? To understand the effects of equity incentive plans implemented by listed companies in China and the potential influence of the general characteristics of contracts on the effectiveness of equity incentive plans. Based on behavioral decision theory, this study adopts a multivariate linear regression model to analyze the 1695 equity incentive plans implemented in Chinese listed companies between 2010 and 2018 with their two-year lagged performance data. The empirical results show that the operational performance of companies after implementing equity incentive plans shows the trend of polarization. In the 95% confidence interval, the effect of restrictive stock incentive and exercise-constrained variables is not significant, while the validity period has a significant positive correlation and incentive intensity has a significantly negative correlation with the company’s operational performance. Furthermore, the negative effects mentioned above become more obvious with a longer plan implementation period. Based on these conclusions, we suggest that companies could adopt equity incentive plans with a relatively longer validity period and more reasonable incentive intensity. Additionally, it would be better for companies to select non-restricted stocks as incentive tools if there is no obvious preference.”

2. No need to have the results in the introduction section.

3. Authors should add the “research structure” for the whole manuscript in the introduction section.

Response to 2 and 3: The research conclusion has been deleted in the introduction and the structure of the paper has been added，Please refer to line 96-104 in tracked version.

“The rest of the paper is arranged as follows: the second part is the review of relevant literature, including the motivation of equity incentive, equity incentive scheme and equity incentive effect; The third part is the proposal of research hypotheses, including the causal effects between equity incentive media and corporate performance, the validity period of equity incentive and corporate performance, and the intensity of equity incentive and corporate performance. The fourth part is empirical research, including research design, descriptive statistics and regression analysis. The fifth part is the discussion and conclusion, including the research conclusion, practical and theoretical significance and future research direction.”

4. Both Introduction and LR parts need to be improved and enriched with extra new papers in the field.

13. Regarding references: most of the references are old. This topic is considered almost as new topic, and many important articles have been published recently in this field. I advise you to refer to previously published articles in Plos One and other top journals to include more updated papers in your study.

Response to 4 and 13: We make some complement，Please refer to line 70-74 , Line175-176, line189-190, line532-535 and so on in tracked version.

line 70-74“For example, study finds that equity incentives are negatively correlated to stock price crash risk in the golden hand cuff regions while equity incentives are positively correlated to stock price crash risk in the golden watch region [3]. In addition, research reveal that high equity incentives could strength CEOs’ degree of earnings management activities [4]”

Line175-176” For example, study reveal that equity incentive exerts positive influence on enterprise’s innovation [26].”

line189-190,” What’s more, study revealed that equity incentive could exert significant influence on corporate innovation performance [30]”

 line532-535” Previous studies mostly discussed the economic consequences from the single variables such as the mode, validity period and intensity of equity incentive. For example, research only select the percentage of shares held by managers as the proxy of equity incentive [26].”

5. Authors need to position their article compared to the academic literature and specify the gap they are trying to fill.

Response: We add new articles to strength this aspect, for example, line70-74, line154-155, Line175-176, line189-190 and so on in tracked version.

line70-74 “For example, study finds that equity incentives are negatively correlated to stock price crash risk in the golden hand cuff regions while equity incentives are positively correlated to stock price crash risk in the golden watch region [3]. In addition, research reveal that high equity incentives could strength CEOs’ degree of earnings management activities [4].”

line 154-155, “Meanwhile, research compared the different effectives among some incentive conditions [20]”

line 175-176, “For example, study reveal that equity incentive exerts positive influence on enterprise’s innovation [26]”

line189-190, “What’s more, study revealed that equity incentive could exert significant influence on corporate innovation performance [30]”

6. The contribution of this paper and the importance of this contribution are not clear; authors should state their contribution and show the importance of this contribution.

8. The authors are not comparing their results with previous work. They are not referring to any reference.

9. The conclusion should be written in paragraphs, not as points.

10. The results and analysis section; it only contains the results with a little discussion. More discussions are needed. Also, the authors should explain more precisely the effect of their results on the Chinese market. In other words, authors should interpret the results beyond their statistical significance to demonstrate their economic impact.

11. Authors should include the "suggestion" section in the conclusion

12. Authors should add the "practical and theoretical implication" and the "limitations and further studies" sections at the end of their paper.

Response to 6,8,9,10,11 and 12: According to the advices, We make some complement，for example, comparing with previous work and explaining the economic impact in line 488-491, line 504-509; We redesign the “conclusion”, in the tracked version, we add Theoretical contribution and practical implications in line 530-560, suggestions in line 561-574, Further Research Directions in line 575-584 and so on in tracked version.

line 488-491, “This conclusion is not only different from extant study that "stock option has a more obvious effect on executive incentive"[40], but also different from research that "equity incentive will increase executive speculation and weaken enterprise value"[2].”

line 502-507, “The research conclusion is consistent with the views of some previous study [40], and is also close to the research results on American enterprises [49]. However, it is just the opposite of study that found a longer term of validity will induce excessive short-term investment behavior, which may be related to the term of the manager [50]. If the manager faces job changes in the short term, the longer term of validity incentive will increase his agency behavior.”

line 528-557, “5.2. Theoretical contribution and practical implications

The marginal theoretical contribution of the paper may be reflected in the following aspects: firstly, the paper enriches the relevant research framework of equity incentive, Previous studies mostly discussed the economic consequences from the single variables such as the mode, validity period and intensity of equity incentive. For example, research only select the percentage of shares held by managers as the proxy of equity incentive [26]. This paper makes a comprehensive study on the above three factors of equity incentive and uses the panel data of Shanghai and Shenzhen listed companies for regression test, so as to obtain more prudent and detailed research conclusions, It also adds more intuitive empirical evidence to verify whether equity incentive intensifying or alleviating the agency phenomenon. 

Secondly, this paper broadens the research on the influencing factors of enterprise performance. In the past, many literatures have discussed the impact of enterprise financial leverage [47]、enterprise risk management [46]、corporate social responsibility [56]、Confucian culture [57]、enterprise power structure [58] and other factors on corporate performance，This paper explores the impact of equity incentive on corporate value and expands the research on the pre variables of enterprise performance.

At the same time, this paper may also have some practical implication, the research conclusion shows that the existing equity incentive model has no significant impact on corporate performance, which shows that in the practice of enterprise management, we should strengthen the innovation of incentive model and explore the equity incentive method suitable for Chinese local enterprises, so as to enhance the practical value of the strategy. In addition, the study found that the longer the validity period of equity incentive, the more it can promote enterprise performance. This conclusion shows that the validity period can be appropriately increased when designing the duration of equity incentive in practice, which helps to maintain the convergence of the interests of management and enterprises. Finally, the study found that higher equity incentive intensity is not conducive to enterprise performance. This conclusion shows that managers holding higher equity share may provide convenience for their self-interest behavior and further aggravate the occurrence of agency phenomenon. Therefore, in practice, we should determine the reasonable range of equity incentive intensity and carefully use the incentive with too high intensity.”

line 561-574, “5.23. Suggestions. The design of incentive mechanism in team is a very important issue, which is related to organizational performance [59]. Based on the research results listed above, this paper offers the following suggestions on the equity incentive plans of listed companies. First, subject matter shows an unapparent negative correlation with company operational performance. Therefore, enterprises should consider their actual situation first when choosing the incentive medium. If there is no preference for any subject matter, companies could consider non-restricted stocks first. Second, enterprises should create reasonable incentive schemes based on their current situation. Based on the significant effect of the validity period and incentive intensity, enterprises can choose a long incentive validity period and they should be more cautious when selecting the intensity of motivation. Last but not the least, equity incentive plans can be an efficient way to test the managerial power of a company. Companies with a “Welfare plan" should pay more attention to improving the internal management structure and reducing management power.”

line 575-584, “5.4. Further Research Directions. This paper discusses the relationship between the types of equity incentive and corporate performance, and it is found that there is no obvious linear relationship between them, In the future research, we can further explore the possible U-shaped and other nonlinear relationship between them. In addition, this paper verifies the significant impact of the validity period and intensity of equity incentive on corporate performance, but the mechanism and boundary of equity incentive have not been discussed in depth, In the future research, we can further analyze the path mechanism of both the impact of equity incentive validity and incentive intensity on corporate performance, as well as the boundary effects such as action intensity in different situations from the perspective of theory and demonstration.”

14. Editing for English is required

Response: owing to the tracked version is too chaotic, meanwhile the paper may confront new change, hence, we’d like to edit the English in the following version.

15. The author mentioned the name of the researchers in addition to the number of the reference in many pages. As in rows of 137, 141, 161, 172, 179, etc

Response: According to the advice, we tracked the above questions, for example, line 152-155 and so on, “Homogeneously, study reached a similar conclusion by discussing small-scale companies, pointing out that a restricted stock mode could weaken agency behavior and reduce agency costs[19]. Meanwhile, research compared the different effectives among some incentive conditions [20].”

7. Methodology: the data used in this research are from 2010 till 2014; these data are outdated. Updated data are required.

Response: According to the advice, new data in paper is from 2010 to 2018, please refer to line 299-307, “implemented in Shanghai and Shenzhen A-share markets between January 1, 2010 and December 31, 2018. Thus, we obtained 1737 equity incentive cases.

In order to ensure the accuracy and validity of the research results further, we further screened the samples and selected them in accordance with the following principles: 

1）Because the different business content of finance and insurance industries leads to different compositions of company assets, liabilities, and owner’s equity, they have an independent accounting system. Therefore, this study excluded financial and insurance companies. 

2）This study excluded invalid values that were missing significant data. 

Following the above steps, 1695 valid samples were obtained.”

And the following statistics tables. For example, table 1 in line 344, table 2 in line 349, ……table 5 in line 467. 

Reviewer 2

Dear expert, we are very grateful for your valuable suggestions, and we will make a specific reply below

1.The structure of the paper should be presented in the introduction.

Response: The structure of the paper has been added，Please refer to line 96-104 in tracked version（all “line” refers to tracked version）. “The rest of the paper is arranged as follows: the second part is the review of relevant literature, including the motivation of equity incentive, equity incentive scheme and equity incentive effect; The third part is the proposal of research hypotheses, including the causal effects between equity incentive media and corporate performance, the validity period of equity incentive and corporate performance, and the intensity of equity incentive and corporate performance. The fourth part is empirical research, including research design, descriptive statistics and regression analysis. The fifth part is the discussion and conclusion, including the research conclusion, practical and theoretical significance and future research direction.”

2. The authors should explain why China is worth studying in relation to other countries.

Response: We make some complement, for example, adding some Chinese study in order to elaborate the complexity of equity incentives in Chinese listed companies, please refer to line 70-75, “For example, study finds that equity incentives are negatively correlated to stock price crash risk in the golden hand cuff regions while equity incentives are positively correlated to stock price crash risk in the golden watch region [3]. In addition, research reveal that high equity incentives could strength CEOs’ degree of earnings management activities [4].”. line 87-92, “Compared with developed capital markets such as Europe, America and Japan, China's equity incentive system is implemented late, the system design is not perfect, and the equity incentive schemes implemented by listed companies are mixed. Therefore, exploring the stock incentive of Chinese listed companies will help to further clarify the economic effect of equity incentive, so as to provide more specific reference value for emerging market countries.”. line 175-176 “For example, study reveal that equity incentive exerts positive influence on enterprise’s innovation [26].” in tracked version. 

3. Tobin Q has been used extensively by papers focused on emerging capital markets. I strongly recommend you to use this measure for robustness checks.

Response: According to the advice, We make a supplement in “Robustness Check” section in line 435-467. “To investigate the robustness of the evaluation methods and conclusions further, this study chose Tobin QROE as an alternative variable to evaluate the operational performance of a company. Similarly, this study constructed the following multivariate linear regression models………”, and the following part

4. You should clarify the contributions of the paper which are not elaborated well in the current paper. You can talk about the following contributions: What insights can you provide based on your finding? Do they push forward our understanding? What should we do with your research? Do you have any suggestions to improve the current regulation or practice? Adding the above discussion and extending your literature review may help you make more contributions and position your contributions better.

The authors should highlight the limits of their research.

Response: According to the advices, We make some complement，for example, comparing with previous work and explaining the economic impact in line 488-491, line 504-509; We redesign the “conclusion”, in the tracked version, we add Theoretical contribution and practical implications in line 530-560, suggestions in line 561-574, Further Research Directions in line 575-584 and so on in tracked version.

line 488-491, “This conclusion is not only different from extant study that "stock option has a more obvious effect on executive incentive"[40], but also different from research that "equity incentive will increase executive speculation and weaken enterprise value"[2].”

line 502-507, “The research conclusion is consistent with the views of some previous study [40], and is also close to the research results on American enterprises [49]. However, it is just the opposite of study that found a longer term of validity will induce excessive short-term investment behavior, which may be related to the term of the manager [50]. If the manager faces job changes in the short term, the longer term of validity incentive will increase his agency behavior.”

line 528-557, “5.2. Theoretical contribution and practical implications

The marginal theoretical contribution of the paper may be reflected in the following aspects: firstly, the paper enriches the relevant research framework of equity incentive, Previous studies mostly discussed the economic consequences from the single variables such as the mode, validity period and intensity of equity incentive. For example, research only select the percentage of shares held by managers as the proxy of equity incentive [26]. This paper makes a comprehensive study on the above three factors of equity incentive and uses the panel data of Shanghai and Shenzhen listed companies for regression test, so as to obtain more prudent and detailed research conclusions, It also adds more intuitive empirical evidence to verify whether equity incentive intensifying or alleviating the agency phenomenon. 

Secondly, this paper broadens the research on the influencing factors of enterprise performance. In the past, many literatures have discussed the impact of enterprise financial leverage [47]、enterprise risk management [46]、corporate social responsibility [56]、Confucian culture [57]、enterprise power structure [58] and other factors on corporate performance，This paper explores the impact of equity incentive on corporate value and expands the research on the pre variables of enterprise performance.

At the same time, this paper may also have some practical implication, the research conclusion shows that the existing equity incentive model has no significant impact on corporate performance, which shows that in the practice of enterprise management, we should strengthen the innovation of incentive model and explore the equity incentive method suitable for Chinese local enterprises, so as to enhance the practical value of the strategy. In addition, the study found that the longer the validity period of equity incentive, the more it can promote enterprise performance. This conclusion shows that the validity period can be appropriately increased when designing the duration of equity incentive in practice, which helps to maintain the convergence of the interests of management and enterprises. Finally, the study found that higher equity incentive intensity is not conducive to enterprise performance. This conclusion shows that managers holding higher equity share may provide convenience for their self-interest behavior and further aggravate the occurrence of agency phenomenon. Therefore, in practice, we should determine the reasonable range of equity incentive intensity and carefully use the incentive with too high intensity.”

line 561-574, “5.23. Suggestions. The design of incentive mechanism in team is a very important issue, which is related to organizational performance [59]. Based on the research results listed above, this paper offers the following suggestions on the equity incentive plans of listed companies. First, subject matter shows an unapparent negative correlation with company operational performance. Therefore, enterprises should consider their actual situation first when choosing the incentive medium. If there is no preference for any subject matter, companies could consider non-restricted stocks first. Second, enterprises should create reasonable incentive schemes based on their current situation. Based on the significant effect of the validity period and incentive intensity, enterprises can choose a long incentive validity period and they should be more cautious when selecting the intensity of motivation. Last but not the least, equity incentive plans can be an efficient way to test the managerial power of a company. Companies with a “Welfare plan" should pay more attention to improving the internal management structure and reducing management power.”

line 575-584, “5.4. Further Research Directions. This paper discusses the relationship between the types of equity incentive and corporate performance, and it is found that there is no obvious linear relationship between them, In the future research, we can further explore the possible U-shaped and other nonlinear relationship between them. In addition, this paper verifies the significant impact of the validity period and intensity of equity incentive on corporate performance, but the mechanism and boundary of equity incentive have not been discussed in depth, In the future research, we can further analyze the path mechanism of both the impact of equity incentive validity and incentive intensity on corporate performance, as well as the boundary effects such as action intensity in different situations from the perspective of theory and demonstration.”

In the end, We thank you very much for your valuable advice, and make above-mentioned modification, we add the two papers (Anton,2018; Anton,2019) in reference 46 and 47.

---

## [Decision Letter · Decision Letter 1]

31 Oct 2022

PONE-D-22-03250R1Equity Incentive Contract Characteristics and Company Operational Performance——An Empirical research of Chinese listed companiesPLOS ONE

Dear Dr. Xu,

Thank you for submitting your manuscript to PLOS ONE. After careful consideration, we feel that it has merit but does not fully meet PLOS ONE’s publication criteria as it currently stands. Therefore, we invite you to submit a revised version of the manuscript that addresses the points raised during the review process.

We look forward to receiving your revised manuscript.

Kind regards,

Haijun Yang, Ph.D

Academic Editor

PLOS ONE

Additional Editor Comments:

Dear Dr. Xu,

After checking the two referees' comments on the article, my decision is major revision. Please verify your manuscript point by point.

Best,

Yang

Reviewers' comments:

Reviewer's Responses to Questions

**Comments to the Author**

1. If the authors have adequately addressed your comments raised in a previous round of review and you feel that this manuscript is now acceptable for publication, you may indicate that here to bypass the “Comments to the Author” section, enter your conflict of interest statement in the “Confidential to Editor” section, and submit your "Accept" recommendation.

Reviewer #2: All comments have been addressed

Reviewer #3: All comments have been addressed

2. Is the manuscript technically sound, and do the data support the conclusions?

Reviewer #2: Yes

Reviewer #3: Yes

3. Has the statistical analysis been performed appropriately and rigorously? 

Reviewer #2: Yes

Reviewer #3: Yes

4. Have the authors made all data underlying the findings in their manuscript fully available?

Reviewer #2: Yes

Reviewer #3: Yes

5. Is the manuscript presented in an intelligible fashion and written in standard English?

Reviewer #2: Yes

Reviewer #3: No

6. Review Comments to the Author

Reviewer #2: Dear Authors,

I have carefully read the revised version of the paper Equity Incentive Contract Characteristics and Company Operational Performance—An Empirical research of Chinese listed companies (PONE-D-22-03250R1) and I checked your responses. I consider that you have improved significantly the paper according to the reviewers’ recommendations. Thus, I consider that the paper can be published in this form. Congrats!

Best regards

Reviewer #3: Dear Author(s)

I would like to thank the author(s) for your submission and appreciate the opportunity to read and review your manuscript. I enjoyed reading it. This manuscript is relevant to contemporary finance research

My comments concern only with clarity, and text. I would recommend publication, after the issues below are addressed.

Changes which must be made before publication:

Major changes:

1. Abstract: : It's okay

2. The Keywords: the Keywords section should deserve more attention, 2-3 keywords should be added.

3. The introduction section: I think it is okay but needs more attention from the author(s).The introduction should give a short summary of your study. The introduction should tell the reader why the study is important, and should seek the attention of the reader by stimulating interest, desire and action).

4. The literature review section: It's okay.

5. Methodology/Research methods: Methodology/Research methods section is missing. Please add a paragraph about the methodology used before the research design.

6. The results section: should be better analyzed and developed further.

7. Conclusions and Further Directions: please augment its quality with more depth, rigor and substance. I recommend you to rewrite this section from p. 27 to p. 31 and avoid use the numeric style in writing this section. This style may be more suitable with reports.

8. The language of the manuscript: the language of the manuscript needs a "Careful Editing" by a native speaker (academic writing style should deserve more attention).

Additional comments: the author(s) are recommended to do the following to increase the manuscript readability.

1. To start with, the manuscript should more clearly and more explicitly spell out its objectives.

2. Please keep your writing in a scientific writing style, the most effective scientific writing aims for accuracy, clarity and objectivity.

3. Please be careful in using the abbreviations, abbreviated words, and definitions throughout the manuscript. Please see p. 16 table 1 the author(s) defined GROW as follows:

• (Profit of this year – Profit of last year) / Profit of last year

I think "GROW" can be defined as follows:

• Net profit growth (%) = (current period NP-Prior Period NP)/Prior Period NP

4. Please be careful in using "Punctuating", e.g. (comma, full stop, etc…) throughout the paper.

5. Some of the terminology used in the manuscript needs to be unified. For instance, the use of 4 different words that have the same meaning interchangeably could confuse some readers such as "paper/ article/research/study".

Finally, based on the above-mentioned comments, I suggest “major revision” of the manuscript before publication.

7. PLOS authors have the option to publish the peer review history of their article (what does this mean?). If published, this will include your full peer review and any attached files.

Reviewer #2: No

Reviewer #3: No

---

## [Author Response · Author response to Decision Letter 1]

16 Dec 2022

Dear Editor and Reviewers,

We are very grateful for your valuable suggestions, and we will make a specific reply below.

1. Abstract: It's okay 

2. The Keywords: the Keywords section should deserve more attention, 2-3 keywords should be added.

Response: As suggested, we add some keywords “Keywords: equity incentive, operational performance, Empirical research, Chinese listed companies, Ownership concentration” in line 30 -31 in tracked version （line 25 in English edited version）. （all “line” refers to tracked version, and “line” in brackets refers to English edited tracked version.）

3. The introduction section: I think it is okay but needs more attention from the author(s). The introduction should give a short summary of your study. The introduction should tell the reader why the study is important, and should seek the attention of the reader by stimulating interest, desire and action).

Response: As suggested, we have made following changes:

(1) Added the manuscript “As an emerging economy with immature capital market and weak corporate governance, how listed companies to design their equity incentive plans would be more helpful to improving operational performance? Answering this question Therefore, exploring the stock incentive of Chinese listed companies will help to further clarify the economic effect of equity incentive” in lines 82-84 ( line 83-87);

(2) Revised “The results of data analysis of 1695 equity incentive cases of China’s listed companies indicate that equity incentive plans are more popular in high-tech industries. Validity periods of incentive plans have significant postive impact on firms’ operational performance, while incentive intensity has a significant negative effect on firms’ operation performance. However, exercise constraints of incentive plans have no significant impact on firms’ operational performance. Our findings have enriched the literature of equity incentive and provide some practical implication for the design of equity incentive plans.” in lines 90-97 (lines 88-96) in order to give a short summary of our study and seek the attention of the reader.

4.The literature review section: It's okay.

5.Methodology/Research methods: Methodology/Research methods section is missing. Please add a paragraph about the methodology used before the research design.

Response: Following this suggestion, we added a new section “Methodology” line 271-261 (line 259-348), and made following modifications:

(1) Move “4.1 research design” to “Methodology” section ( line 290-346 in former edition of manuscript), as subsection “4.1. Sample and sources of data” in line 275-292 (lin 266-282) and subsection “4.2. Variable definition” in line 293-318 (283-319);

(2) Move model specification part in “4.2.2. Results of Multiple linear regressions” ( line 364-384 in former edition of manuscript) to “Methodology” section as a subsection “4.3. Model specification”, in line 329-367 ( line 320-354);

(3) All contents mentioned above were revised to fit to the new section.

6. The results section: should be better analyzed and developed further.

Response: Following the suggestion, we have revised the results section to make it more accurate and academic style. Major changes are as follows:

(1) “A comparison of the three-year ROA data after implementation shows that the mean value of ROA is declining, but the max value is increasing, and it shows a growing polarization trend in the three years. Therefore, the operational performance of listed companies who have implemented equity incentive plans experienced large-scale variance, which means that the impact of stock incentive plans with different characteristics might be different.” In line 374-380 ( line 361-365);

(2) “Pearson correlation coefficients for each variable are presented in Table 3. It is worth noting that there was a negative correlation between STOCK and the other variables. Because STOCK is a virtual variable and 1 represents restricted stocks while 0 represents non-restricted stocks, it can be found that restricted stock has an inhibitory effect on other explanatory and control variables. By comparing all Pearson correlation indices between variables, it can be found that all value of correlation coefficients are below 0.4 (a few indices are around 0.4). Hence, the correlations between these variables are weak. Therefore, there should have no multiple co-linear problems between these variables in the regression models.” In line 405-415 ( line 374-381);

(3) “From the results of the regression model shown above, it can be seen that the adjusted coefficients of determination are above 0.25, which means that these models have strong explanatory ability. A comparison of the last four models (Models (6)–(8)) shows that Model (6)’s goodness of fit is the best, Model (7) is the second best, and Model (8) is the worst. The results shows that all the regression models are significant at the 0.001 level.

…

The above results indicate that, an equity incentive plan with a long validity period can greatly promote a company’s operational performance, and the impact reaches its peak in the second year after equity incentive implementation. However, incentive intensity has an negative effect on a company’s operational performance.” in line 425-468 ( line 390-427).

7.Conclusions and Further Directions: please augment its quality with more depth, rigor and substance. I recommend you to rewrite this section from p. 27 to p. 31 and avoid use the numeric style in writing this section. This style may be more suitable with reports.

Response: As suggested, we have dropped the numeric style and made some supplement, such as:

“Undoubtedly, senior executives of high-tech enterprises usually have higher professional skills, and such professional skills have a higher threshold. The above factors determine that it is necessary to implement appropriate incentives.” in line 507-509 ( line 448-450);

“they used China's GEM listed companies as a sample, this kind of listed companies started late, have better adaptability and flexibility, and usually do not have the inertia of hard to return” in line 515-517 ( line 456- 459);

“Chinese officials are trying to avoid corporate fraud, but due to the characteristics of emerging markets in the transition period, Chinese enterprises still face the situation of senior executives cheating in the implementation of equity incentive, which is just like the situation in the western capital markets in the past” in line 519-522 ( line 460-464);

“undertake many social functions and the senior executives‘ special status,” in line 527 ( line 469);

“The Chinese economy has experienced rapid growth and development these years, however, the system designed for corporate governance lags behind.” in line 529-531 ( line 471-473);

“some Chinese scholars believe that executives with relatively short incentive and constraint time may prefer earnings management, stock price manipulation and other methods to meet the exercise conditions; meanwhile, for a long time of incentive and constraint, senior executives are more likely to truly create enterprise value locally by improving operating efficiency, and is also close to the research results on American enterprises, for example, scholars select 1932 firms as sample and reveal that long-term incentives have a stronger inﬂuence on innovation” in line 538-544 ( line 479-486);

“The managers have incentives to overinvest in short-term projects to boost the board’s attitude about their abilities, especially” in line 546-547 ( line 489-490);

“just as we know that the US capital market is more perfect than the UK, hence, the US management has a lower level of ownership than their British counterparts, which could strengthen the positive effect between equity incentive and corporate performance” in line 555-558 ( line 497-500);

“For example, the power of senior executives can ensure the self-serving behavior of senior executives. The greater the power of senior executives, the greater the intensity of equity incentive; Meanwhile, the announcement of the draft equity incentive will produce a significantly positive market response, and the market can be aware of the self-serving behavior of senior executives.” in line 559-562 ( line 501-506);

“the study of S&P 500 ﬁrms in the1990s suggests the market value of the company is improved when the pay for performance plan directly for the company's executives is disclosed, the effect of implementing this plan is obvious in companies with potential agency, because this stock compensation mechanism has stimulated the efforts of executives and led to the improvement of company performance.” in line 566-570 ( line 509-513);

“On the contrary, the sample of S&P companies in European and American capital markets is more mature, which means more reasonable policy supervision and market expectations” in line 572-574 ( line 515-517);

“In spite of the aforementioned theoretical significance and practical enlightenment, however, there are still some imperfections in this paper.” in line 629-631 ( line 575-577).

8.The language of the manuscript: the language of the manuscript needs a "Careful Editing" by a native speaker (academic writing style should deserve more attention).

Response: As suggested, we have looked MDPI company for help, they arranged for a senior editor to deal with language problems.

Additional comments: the author(s) are recommended to do the following to increase the manuscript readability.

1. To start with, the manuscript should more clearly and more explicitly spell out its objectives.

Response: As suggested, we have revised the “instruction” in line 82-97 ( line 83-96):

“As an emerging economy with immature capital market and weak corporate governance, how listed companies to design their equity incentive plans would be more helpful to improving operational performance? Answering this question Therefore, exploring the stock incentive of Chinese listed companies will help to further clarify the economic effect of equity incentive, so as to provide more specific reference value for public firms in emerging market countries.

Therefore, Based on the above considerations, this paper study focuses on the impact of design factors such as equity incentive mode, effective period, and other endogenous conditions on the performance of plan implementation and proposes suggestions on the design of equity incentive plans based on the Chinese listed companies. The results of data analysis of 1695 equity incentive cases of China’s listed companies indicate that equity incentive plans are more popular in high-tech industries. Validity periods of incentive plans have significant positive impact on firms’ operational performance, while incentive intensity has a significant negative effect on firms’ operation performance. However, exercise constraints of incentive plans have no significant impact on firms’ operational performance. Our findings have enriched the literature of equity incentive and provide some practical implication for the design of equity incentive plans.”

2. Please keep your writing in a scientific writing style, the most effective scientific writing aims for accuracy, clarity and objectivity.

Response: As suggested, we have made language editing, in addition, we also revised the instruction, main body, especially the conclusion part.

3. Please be careful in using the abbreviations, abbreviated words, and definitions throughout the manuscript. Please see p. 16 table 1 the author(s) defined GROW as follows:

• (Profit of this year – Profit of last year) / Profit of last year

I think "GROW" can be defined as follows:

• Net profit growth (%) = (current period NP-Prior Period NP)/Prior Period NP

Response: we have revised in page 17 (table 1) line 328 (line 319)

“Net profit growth (%) = (Current period NP – Prior period NP) / Prior period NP”

4. Please be careful in using "Punctuating", e.g. (comma, full stop, etc…) throughout the paper.

Response: As suggested, we have made several changes throughout the paper, together with English language edit.

5.Some of the terminology used in the manuscript needs to be unified. For instance, the use of 4 different words that have the same meaning interchangeably could confuse some readers such as "paper/ article/research/study".

Response: As suggested, we have made several changes throughout the paper. Such as line 18, line 63, line 87, line 118, line 137, line 149, line 159, line 172……, and so on.

---

## [Decision Letter · Decision Letter 2]

19 Jan 2023

Equity Incentive Contract Characteristics and Company Operational Performance - An Empirical Study of Chinese Listed Companies

PONE-D-22-03250R2

Dear Dr. Xu,

We’re pleased to inform you that your manuscript has been judged scientifically suitable for publication and will be formally accepted for publication once it meets all outstanding technical requirements.

Kind regards,

Haijun Yang, Ph.D

Academic Editor

PLOS ONE

Additional Editor Comments (optional):

Dear Dr. Xu,

The reviewers consider that you has improved significantly the paper according to the reviewers’ recommendations. My decision is accept.

Best,

Yang

Reviewers' comments:

Reviewer's Responses to Questions

**Comments to the Author**

1. If the authors have adequately addressed your comments raised in a previous round of review and you feel that this manuscript is now acceptable for publication, you may indicate that here to bypass the “Comments to the Author” section, enter your conflict of interest statement in the “Confidential to Editor” section, and submit your "Accept" recommendation.

Reviewer #2: All comments have been addressed

Reviewer #3: All comments have been addressed

2. Is the manuscript technically sound, and do the data support the conclusions?

Reviewer #2: Yes

Reviewer #3: Yes

3. Has the statistical analysis been performed appropriately and rigorously? 

Reviewer #2: Yes

Reviewer #3: Yes

4. Have the authors made all data underlying the findings in their manuscript fully available?

Reviewer #2: Yes

Reviewer #3: Yes

5. Is the manuscript presented in an intelligible fashion and written in standard English?

Reviewer #2: Yes

Reviewer #3: Yes

6. Review Comments to the Author

Reviewer #2: Dear Authors,

I have carefully read the revised version of the paper Equity Incentive Contract Characteristics and Company Operational Performance——An Empirical research of Chinese listed companies (PONE-D-22-03250R2) and I checked your responses. I consider that you have improved significantly the paper according to the reviewers’ recommendations. Thus, I consider that the paper can be published in this form. Congrats!

Best regards

Reviewer #3: Dear Author(s)

I would like to thank the author(s) for addressing my initial comments. The author(s) have greatly improved their manuscript and responded to the points that I have raised. Following the revision to the paper, additional "minor comment" relate to the amendments made. My remaining comment concern only with the logic of the research. I would recommend publication, after the issue below are addressed.

Minor changes:

The methodology section: Under the sub-section 4.2. Variable definition (1) dependent variables p. 14 line 287 - 292.

The author(s) mentioned that "Instead, Chinese scholars often use the return on total assets (ROA) in similar studies to measure companies’ operational performance. For example, researchers explored the impact of equity incentive schemes on company operational performance using the rate of return on equity (ROE), the rate of return on total assets (ROA), and the main business profit (ROM) as dependent variables [45]. Based on the abovementioned studies, we selected the return on total assets (ROA) as the dependent variable."

From my point of view and to avoid confusing the readers, in fact the (ROE) and the (ROA) Measuring the firm's operational performance or its "capital productivity" while the (ROM) "Return-on-Management" measuring its "management productivity". In view of the fact that your research focus on the operational performance, I suggest to rewrite this paragraph and please avoid mentioned the (ROM).

Finally, based on the above-mentioned comment, I suggest “minor revision” of the paper before publication.

7. PLOS authors have the option to publish the peer review history of their article (what does this mean?). If published, this will include your full peer review and any attached files.

Reviewer #2: No

Reviewer #3: No

---

## [Editor Report · Acceptance letter]

26 Jan 2023

PONE-D-22-03250R2 

Equity Incentive Contract Characteristics and Company Operational Performance - An Empirical Study of Chinese Listed Companies 

Dear Dr. Xu:

I'm pleased to inform you that your manuscript has been deemed suitable for publication in PLOS ONE. Congratulations! Your manuscript is now with our production department. 

Kind regards, 

on behalf of

Dr. Haijun Yang 

Academic Editor

PLOS ONE